# Multi-Fidelity Best-Arm Identification

**Riccardo Poiani**
DEIB, Politecnico di Milano
riccardo.poiani@polimi.it

**Alberto Maria Metelli**
DEIB, Politecnico di Milano
albertomaria.metelli@polimi.it

**Marcello Restelli**
DEIB, Politecnico di Milano
marcello.restelli@polimi.it

## Abstract

In several real-world applications, a learner has access to multiple environment simulators, each with a different precision (e.g., simulation accuracy) and cost (e.g., computational time). In such a scenario, the learner faces the trade-off between selecting expensive accurate simulators or preferring cheap imprecise ones. We formalize this setting as a multi-fidelity variant of the stochastic best-arm identification problem, where querying the original arm is expensive, but multiple and biased approximations (i.e., fidelities) are available at lower costs. The learner's goal, in this setting, is to sequentially choose which simulator to query in order to minimize the total cost, while guaranteeing to identify the optimal arm with high probability. We first derive a lower bound on the identification cost, assuming that the maximum bias of each fidelity is known to the learner. Then, we propose a novel algorithm, Iterative Imprecise Successive Elimination (IISE), which provably reduces the total cost w.r.t. algorithms that ignore the multi-fidelity structure and whose cost complexity upper bound mimics the structure of the lower bound. Furthermore, we show that the cost complexity of IISE can be further reduced when the agent has access to a more fine-grained knowledge of the error introduced by the approximators. Finally, we numerically validate IISE, showing the benefits of our method in simulated domains.

## 1 Introduction

In the multi-armed bandit setting [26], the agent selects, at each interaction round, an arm and observes a sample drawn from its reward distribution. Unlike the regret minimization setting [26], where the agent tries to maximize the cumulative return, in *best-arm identification* (BAI) problems [10, 6, 2], the evaluation index relates exclusively to the quality of the arm recommended at the end of the exploration phase. More specifically, in the *fixed-confidence* setting [17], the agent aims at identifying, with probability $1 - \delta$, the arm with the largest mean reward, while minimizing the sample complexity, i.e., the number of collected samples necessary to make the final decision. Due to its generality, the BAI framework encompasses a wide spectrum of applications, ranging from medical trials [38], to hyper-parameter optimization [27], crowdsourcing [47], recommender systems [25], simulation-based studies in physics [43], and A/B testing [14].

However, in practical cases, querying an arm to obtain a sample from its reward distribution might be expensive. For instance, in the context of physics simulation studies, where pulling an arm corresponds to the evaluation of a complex model under some given arm parameters, intensive use of computing power is required to obtain the reward we are interested in. Nevertheless, simpler and, consequently, cheaper and biased, models might be available to the agent. Another example is stochastic planning, where the agent needs to compute the best action (i.e., arm) that maximizes the future sum of rewards over a maximum horizon $H$. However, while in principle it is possible to plan

36th Conference on Neural Information Processing Systems (NeurIPS 2022).

considering the entire horizon, cheaper and biased estimates can be obtained by cutting the depth of the search to $h < H$. For these scenarios, the *multi-fidelity* (MF) framework [9, 21, 22] that exploits this additional opportunity has recently gained attention in the literature. The main rationale behind these methodologies is to exploit less expensive approximations in specific regions of the arm space to limit the resources allocated to distinguish options that are far from optimal, and do not require high accuracy. So far, most of the multi-fidelity literature has focussed on black-box function optimization, with the goal of optimizing a target function $f$ when multiple approximations of $f$ are available [15, 33, 22, 34, 39, 40, 19, 28, 11]. Among these works, the ones more related to the bandit literature are [22, 19, 39, 40, 11]; under different smoothness assumptions, these authors aimed at designing strategies that explore the arm space to minimize the *simple regret*, i.e., the difference between the true optimal value of $f$ and the best unbiased query made to $f$ during the optimization process. Other works studied cumulative regret analysis [21, 20], active learning [46, 32], reinforcement learning [9, 30, 41], and applied approaches [36, 44, 31].

In this paper, on the other hand, we focus on the understudied multi-fidelity fixed-confidence BAI setting. More specifically, we study the minimization of the *cost complexity*, a novel metric that we introduce by extending the notion of sample complexity to the multi-fidelity setting, by weighting each sample with the related cost. The goal of the agent is to output the optimal arm with probability $1 - \delta$, while minimizing the total cost. To this purpose, as commonly done in the literature [9, 21, 20, 39, 11], we assume that the agent has some knowledge of the quality of the approximator, which is usually expressed in terms of upper bounds on the maximum bias that the fidelities introduce on the arm set. This term corresponds to the maximum distance between the mean of the desired (but expensive) arm distributions and their approximations.

**Contributions.** In this paper, we introduce the notion of cost complexity for the multi-fidelity BAI setting (Section 2). We derive a *lower bound* for this novel index, under the standard assumption that the agent is equipped with knowledge of the maximum bias that each fidelity has w.r.t. the original, expensive and unbiased bandit model (Section 3). As we shall show, our result highlights several properties of the problem along with its challenges. Furthermore, we propose the Iterative Imprecise Successive Elimination (IISE) algorithm (Section 4), for which, we derive cost complexity upper bounds that, under certain assumptions, mimic the structure of the lower bound, and that is provably robust w.r.t. BAI algorithms that ignore the multi-fidelity structure. We also show that the performance of IISE can be further improved when a more fine-grained knowledge of the behavior of the approximations is available to the agent (Section 5). After revising the literature (Section 6), we propose numerical experiments on simulated domains that empirically verify our claims (Section 7).

## 2 Setting

In the classical *fixed-confidence BAI* setting [17], the agent interacts with a bandit model $\nu := (\nu_1, \ldots, \nu_K)$, where $K \in \mathbb{N}$ is the number of arms and $\nu_i$ denotes the reward distribution associated with arm $i \in [K] := \{1, \ldots, K\}$. Denote with $\mu_i$ the expectation of $\nu_i$ and, assume, that the $K$ arms are ordered such that $\mu_1 > \mu_2 \geq \cdots \geq \mu_K$. We assume that the reward distributions $\nu_i$ are subgaussian with known scale parameter $\sigma^2$. At each interaction round $t \in \mathbb{N}$, the agent selects an arm $I_t \in [K]$ and receives a reward $R_t$ sampled from $\nu_{I_t}$. Let $T_i(t)$ be the total number of pulls of arm $i \in [K]$ up to time $t \in \mathbb{N}$. Given a confidence level $\delta \in (0, 1)$, the goal of the agent is to recommend an arm $\widehat{I}(\tau) \in [K]$ that corresponds to the optimal arm 1 with probability at least $1 - \delta$, while minimizing the sample complexity $\rho(\tau) := \sum_{i \in [K]} \mathbb{E}[T_i(\tau)]$, where $\tau$ is the stopping time of the agent. Any algorithm that recommends the best arm with probability $1 - \delta$ is said $\delta$-correct. For every sub-optimal arm $i > 1$, we define its gap w.r.t. the optimal arm as $\Delta_i := \mu_1 - \mu_i$.

In this paper, we differ from this classical setting as, for each arm $i \in [K]$, we assume the agent has access to $M - 1 \in \mathbb{N}$ fidelity distributions $\nu_{i,1}, \nu_{i,2}, \ldots, \nu_{i,M-1}$, each $\sigma^2$-subgaussian, with means $\mu_{i,1}, \mu_{i,2}, \ldots, \mu_{i,M-1}$. We unify the notation introducing a maximum fidelity $M$ to encode the true arm distributions the agent is interested in, i.e., $\nu_{i,M} := \nu_i$ with mean $\mu_{i,M} := \mu_i$. As in previous works [9, 21, 20, 39, 11], we assume that there is a relation encoding how much information distributions of fidelity $m < M$ contain w.r.t. to the true arm distribution. More specifically, for each *fidelity* $m \in [M]$, it is always possible to write $\mu_{i,m} := \mu_{i,M} + \xi_{i,m}$ for some appropriate $\xi_{i,m} \in \mathbb{R}$. Then, we assume that the agent has access to $\xi_m \geq \max_{i \in [K]} |\xi_{i,m}|$, from which follows that $|\mu_{i,m} - \mu_{i,M}| \leq \xi_m$. We notice that $\xi_m$ represents an *upper-bound on the maximum bias* of fidelity $m \in [M]$. At each

interaction round $t \in \mathbb{N}$, the agent selects an arm $I_t \in [K]$, a fidelity $m_t \in [M]$, and observes a reward $R_t$, drawn from $\nu_{I_t, m_t}$. The cost of gathering samples at fidelity $m$ is known and specified by $\lambda_m > 0$. We make the standard assumption that $\lambda_1 < \lambda_2 < \cdots < \lambda_M$ and $\xi_1 > \xi_2 > \cdots > \xi_M := 0$.[1] At this point, let $T_{i,m}(t)$ be the number of pulls to arm $i \in [K]$ at fidelity $m \in [M]$ up to time $t \in \mathbb{N}$; we define the *cost complexity* as:

$$c(\tau) := \sum_{i \in [K]} \sum_{m \in [M]} \lambda_m \mathbb{E}\left[ T_{i,m}(\tau) \right], \tag{1}$$

where $\tau$ is the stopping time at which the agent will output the optimal arm 1 with probability $1 - \delta$.

## 3 Cost Complexity Lower Bound

In this section, we discuss the intrinsic complexity of the multi-fidelity BAI problem, by providing and analyzing a lower bound on the cost complexity (proof in Appendix B).

**Theorem 1.** *Consider a multi-fidelity bandit model $\nu$ with Gaussian distributions $\nu_{i,m} = \mathcal{N}(\mu_{i,m}, \sigma^2)$ such that $|\mu_{i,m} - \mu_{i,M}| \le \xi_m$ for every $i \in [K]$ and $m \in [M]$. Then, for any $\delta$-correct algorithm and $\delta \le 0.15$, it holds that:*

$$\mathbb{E}\left[c(\tau)\right] \ge \left[ \min_{m \in \mathcal{M}_1} \frac{\lambda_m}{\mathrm{KL}(\nu_{1,m}, \overline{\nu}_{2,m})} + \sum_{i=2}^{K} \min_{m \in \mathcal{M}_i} \frac{\lambda_m}{\mathrm{KL}(\nu_{i,m}, \overline{\nu}_{1,m})} \right] \log\left( \frac{1}{2.4\delta} \right),$$

*where $\mathrm{KL}(p, q)$ is the Kullback-Leibler divergence between distributions $p$ and $q$, $\overline{\nu}_{2,m} = \mathcal{N}(\mu_{2,M} + \xi_m, \sigma^2)$, $\overline{\nu}_{1,m} = \mathcal{N}(\mu_{1,M} - \xi_m, \sigma^2)$, $\mathcal{M}_1 := \{m \in [M] : \mu_{1,m} > \mathbb{E}_{x \sim \overline{\nu}_{2,m}}[x]\}$ and $\mathcal{M}_i := \{m \in [M] : \mathbb{E}_{x \sim \overline{\nu}_{1,m}}[x] > \mu_{i,m}\}$ for $i > 1$.*

This bound reveals several properties of the multi-fidelity fixed-confidence BAI problem. A single term $\lambda_m \mathrm{KL}^{-1}(\nu_{i,m}, \overline{\nu}_{1,m})$ with $i \ne 1$ can be interpreted as a lower bound on the minimum cost that any $\delta$-correct algorithm needs to pay to conclude that arm $i$ is sub-optimal. Suppose, for the moment, that $\mathcal{M}_i$ contains any fidelity $m \in [M]$. Then, once we fix $m \in \mathcal{M}_i$, its related component $\lambda_m \mathrm{KL}^{-1}(\nu_{i,m}, \overline{\nu}_{1,m})$ is the cost-based multi-fidelity counterpart of the usual $\mathrm{KL}^{-1}(\nu_i, \nu_1)$ which appears in the standard BAI lower bound [24]. This can be seen as a lower bound on the cost to discard arm $i$ using samples gathered at fidelity $m$. Notice that, in this sense, the term $\overline{\nu}_{1,m}$ replaces the usual $\nu_1$ to take the bias of fidelity $m$ into account. The minimum, consequently, encodes a lower bound on the cost of the *most convenient* fidelity with which we can discard $i$. The reason why we have a minimum over $\mathcal{M}_i$ instead of $[M]$ is that, given a fidelity $m \in [M]$, $m$ might not be useful to discard $i$, since $\Delta_i$ might be too small for the bias that $m$ introduces. This concept is further clarified in the following remark:

**Remark 1.** *Consider the following problem instance: for each $m \in [M]$, let $\mu_{1,m} = \mu_{1,M} - \xi_m$ and $\mu_{i,m} = \mu_{i,M} + \xi_m$ for $i \ne 1$. Then, Theroem 1 reduces to:*

$$\widetilde{\Omega}\left( \sum_{i=2}^{K} \min_{m \in [M] : \Delta_i > 2\xi_m} \frac{\lambda_m \sigma^2}{(\Delta_i - 2\xi_m)^2} \right). \tag{2}$$

As we can see, if $\Delta_i < 2\xi_m$ for some $m$, its related component is not present in the lower bound. Furthermore, it is interesting to note that the setting of Remark 1 represents a worst-case fidelity-dependent scenario. Indeed, each fidelity $m \in [M-1]$ underestimates the reward of the optimal arm and overestimates the ones of sub-optimal arms.

Finally, we show how Theorem 1 highlights the main challenge of our setting. For each value of $\xi_m$, there exists a multi-fidelity bandit model $\nu$ for which the lower bound is given by $\widetilde{\Omega}(\sum_{i=2}^{K} \frac{\lambda_M \sigma^2}{\Delta_i^2})$ (given Remark 1, it is sufficient to consider $\Delta_i$ such that the only fidelity in the minimization set is $M$). It is clear that this cost complexity can be matched (up to log factors) by many BAI algorithms available in the literature (e.g., [10]) that ignore the multi-fidelity structure of the problem. The main

---

[1]Cases in which $\lambda_i = \lambda_{i+1}$ or $\xi_i = \xi_{i+1}$ hold can easily be ruled out. Indeed, with the same cost one could gather a more precise sample using another fidelity, or, equivalently, one could gather a sample of the same precision, using a cheaper fidelity; i.e., we can remove these dominated fidelity prior to running any method.

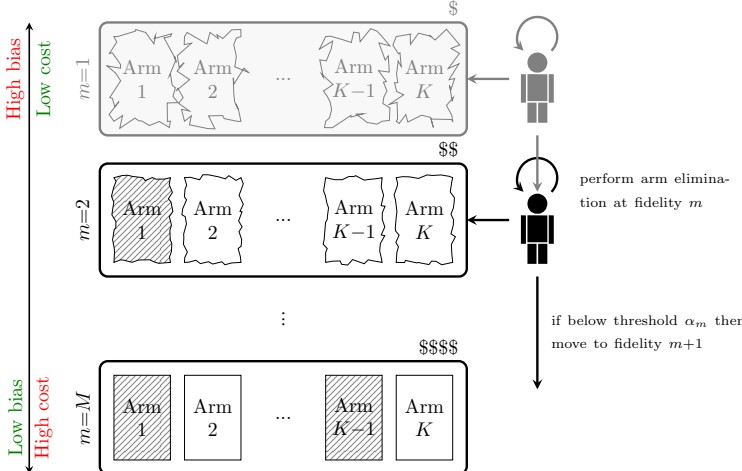

Figure 1: A visualization of how IISE works: at the beginning IISE pulls arms at the cheapest fidelity in a Successive Elimination fashion. Using samples from these biased distributions, it tries to discard sub-optimal $i$ arms whose $\Delta_i$ is sufficiently large. Once a given number of samples determined by $\alpha_m$ has been reached, IISE moves to the sub-sequent fidelity. This process goes on until the optimal arm is identified in high-probability.

issue, in this sense, is the fact that the agent does not know, a priori, the values of $\Delta_i$. To build an algorithm that achieves *robustness* (i.e., not being worse than these uninformed strategies), then, we need to control how much we spend to explore the arm distributions at fidelity $m < M$. More generally, this reasoning also extends to building algorithms that match the lower bound. In this sense, we observe, even for the fixed-confidence BAI setting, the adversarial behavior that fidelity might have, as previously reported in [21] for the finite-armed bandit regret minimization setting.

## 4 Imprecise and Iterative Successive Elimination

In this section, we present our algorithm, *Imprecise and Iterative Successive Elimination* (IISE, Algorithm **??**), to solve multi-fidelity fixed-confidence BAI problems. In particular, while developing our method, we pursue the following two objectives. First, we want to guarantee *robustness* w.r.t. uninformed strategies that blindly use only the highest fidelity $M$ to identify the best arm. Indeed, we expect a good agent to be able to reduce the total cost by exploiting the peculiar multi-fidelity structure of the problem. Second, we aim at providing a strategy whose upper-bound structure resembles, as close as possible, the one of the lower bound presented in Section 3.

Our solution (pseudo-code in Algorithm 1 and visual representation in Figure 1) builds on the Successive Elimination algorithm [10]. More specifically, we proceed by epochs, one for each fidelity $m \in [M]$, considered in increasing order. In each phase $m \in [M]$, we query all the *active* arms, i.e., those that have not been discarded yet, through their approximated distributions $\nu_{i,m}$ at fidelity $m$. We denote the set of active arms with $S \subseteq [K]$. IISE keeps usual empirical mean estimates $\hat{\mu}_{i,m,t}$ of the mean $\mu_{i,m}$ of reward distributions $\nu_{i,m}$ for each $i \in S$, and discards all the arms $i$ for which the following applies:

$$\exists j \in S: \qquad \hat{\mu}_{j,m,t} - U\left(t, \frac{\delta}{KM}, \xi_m\right) \geq \hat{\mu}_{i,m,t} + U\left(t, \frac{\delta}{KM}, \xi_m\right), \qquad (3)$$

where $U(t, \delta, \xi)$ is a symmetric bound on the empirical means (defined below). Condition (3) aims at eliminating all the active arms $i$ whose estimated empirical mean upper bound falls below the empirical mean lower bound of another active arm $j$. More specifically, we set $U(t, \delta, \xi) \coloneqq B(t, \delta) + \xi = \sqrt{\frac{2\sigma^2 \log \frac{4t^2}{\delta}}{t}} + \xi$. As we can see, $U(t, \delta, \xi)$ is composed of the usual concentration term $B(t, \delta)$ plus the fidelity bias $\xi$. As our proof highlights, this additional term is required to

---

**Algorithm 1** Iterative Imprecise Successive Elimination (IISE).

---

**Require:** Multi-fidelity bandit model $\nu$, confidence $\delta$, thresholds $\{\alpha_m\}_{m=1}^M$, bounds $\{\xi_m\}_{m=1}^M$

1: $S \leftarrow [K]$
2: $m \leftarrow 1$ and $t \leftarrow 0$
3: **while** $|S| > 1$ **do**
4:     **if** $\alpha_m \geq 4B\left(t, \frac{\delta}{KM}\right)$ **then**
5:         $m \leftarrow m+1$ and $t \leftarrow 0$
6:     **end if**
7:     Pull all arms in $S$ at fidelity $m$ and $t \leftarrow t+1$
8:     Update $\hat{\mu}_{j,m,t}$ for all $j \in [S]$
9:     $S \leftarrow S \setminus \left\{ i \in S : \exists j \in S : \hat{\mu}_{j,m,t} - U\left(t, \frac{\delta}{KM}, \xi_m\right) \geq \hat{\mu}_{i,m,t} + U\left(t, \frac{\delta}{KM}, \xi_m\right) \right\}$
10: **end while**
11: **return** $S$

---

ensure that the optimal arm remains, with high probability, in the active set $S$ until termination.[2] The duration of each epoch (i.e., how much IISE tries to exploit fidelity $m$) is determined by a *threshold* parameter $\alpha_m$ (to be defined later). At each round, the value of $B\left(t, \frac{\delta}{KM}\right)$ decreases and as soon as $4B\left(t, \frac{\delta}{KM}\right)$ drops below $\alpha_m$, IISE switches to the next fidelity $m+1$. Notice that, to achieve $\delta$-correctness, it is required to set $\alpha_M$ to 0, so that the last phase is guaranteed to identify the optimal arm, and $\alpha_m > 0$ for $m < M$, so that each phase is assured to terminate in a finite number of rounds. Then, keeping generic thresholds $\alpha_m$ that satisfy the previous conditions, we can show that IISE is $\delta$-correct and its cost complexity is upper bounded by the following theorem (proof in AppendixC).

**Theorem 2.** *If $\alpha_m > 0$ for every $m \in [M-1]$ and $\alpha_M = 0$, then, with probability at least $1 - \delta$, IISE returns the optimal arm $1$ with cost complexity $c(\tau)$ upper bounded by:*

$$c(\tau) \leq O\left( \sum_{i=2}^K \frac{\lambda_{m_i}\sigma^2}{(\Delta_i - 4\xi_{m_i})^2} \log\left(\frac{\sigma^2 MK}{(\Delta_i - 4\xi_{m_i})^2\delta}\right) + \sum_{m < m_i} \frac{\lambda_m \sigma^2}{\alpha_m^2} \log\left(\frac{\sigma^2 MK}{\alpha_m^2 \delta}\right) \right),$$

*where $m_i$ is the smallest $m \in [M]$ for which $\Delta_i \geq 4\xi_m + \alpha_m$ holds.*

We notice that the cost required by IISE to discard $i \neq 1$ is composed of two terms. The first one is the cost of discarding arm $i$ using samples at fidelity $m_i$, while the second one is the cost of trying to discard arm at lower fidelity $m < m_i$. As our proof reveals, $m_i$ is the smallest fidelity $m$ for which IISE ensures that arm $i$ will be discarded.

## 4.1 On the Choice of the Thresholds $\alpha_m$

In this section, we highlight one of the main challenges of the MF setting, as mentioned in Section 3. Since we are unaware of the suboptimality gaps $\Delta_i$, we may attempt to choose $\alpha_m$ so as to minimize the cost complexity of Theorem 2 w.r.t. the worst-case choice of suboptimality gaps $\Delta_i$, that is:

$$\min_{\boldsymbol{\alpha}} \max_{\boldsymbol{\Delta}} \sum_{i=2}^K \frac{\lambda_{m_i}\sigma^2}{(\Delta_i - 4\xi_{m_i})^2} \log\left(\frac{\sigma^2 MK}{(\Delta_i - 4\xi_{m_i})^2\delta}\right) + \sum_{m < m_i} \frac{\lambda_m \sigma^2}{\alpha_m^2} \log\left(\frac{\sigma^2 MK}{\alpha_m^2 \delta}\right), \quad (4)$$

where $\boldsymbol{\alpha} = (\alpha_1, \ldots, \alpha_M)$ and $\boldsymbol{\Delta} = (\Delta_2, \ldots, \Delta_K)$ are the threshold and gap vectors. However, it is easy to verify that Equation (4) is minimized for $\alpha_m = +\infty$ for all $m < M$ and $\alpha_M = 0$, i.e., an algorithm that exploits the highest fidelity $M$ only. As we can see, IISE reflects the peculiar issue of the MF-BAI lower bound. In the worst case, any fidelity $m < M$ is not useful to reduce the identification cost since it introduces too much bias w.r.t. $\Delta_i$, regardless its cost. For this reason, as done in previous works [21], we restrict the class of problems with some assumptions whose goal is making the fidelity structure provably convenient. More specifically:

**Assumption 1** (Costs and Biases Relationship). *For every fidelity $\overline{m} \in [M-1]$, it holds that:*

$$\sum_{m < \overline{m}} \min_{k > m} \frac{(\sqrt{\lambda_k} - \sqrt{\lambda_m})^2}{(\xi_m - \xi_k)^2} \leq \min_{k > \overline{m}} \frac{(\sqrt{\lambda_k} - \sqrt{\lambda_{\overline{m}}})^2}{(\xi_{\overline{m}} - \xi_k)^2} \quad (5)$$

---

[2]Omitting $\xi$ in the elimination rule might wrongly eliminate the optimal arm just because, at low fidelity, its approximated distribution has a negative bias.

Assumption 1 deserves some comments. First of all, it directly encodes a relationship between costs and biases. To provide an interpretation, we notice that the assumption is always satisfied for: (i) costs that grow at least linearly [3] and (ii) $\xi$'s that satisfy the following decay rate: $\sum_{m<\overline{m}} \frac{1}{(\xi_m - \xi_{m+1})^2} \leq \frac{1}{\xi_{\overline{m}}^2}$. The intuition behind the cost increase rate (i) is that, with sub-linear growth, attempting to discard arms with a cheaper fidelity is too inconvenient w.r.t. gathering more accurate samples for a slightly higher cost. For what concerns the $\xi$'s decay rate, instead, condition (ii) limits the adversarial behavior that fidelity might have. If two subsequent $\xi$'s are very close together, little can be inferred from switching from the first fidelity to the second one. This sort of problem was already highlighted in [21]. We remark that Assumption 1, compared to Assumption 1 of [21] $\sum_{m<\overline{m}} \frac{1}{(\xi_m)^2} \leq \frac{1}{\xi_{\overline{m}}^2}$, involves both costs and biases, which is something we expect since the main challenge of the setting is to trade-off between them. Indeed, depending on the values of $\lambda$'s it is possible to tolerate $\xi$'s that decay linearly; vice-versa, if $\xi$'s decay fast enough, it is possible to tolerate costs that increase sub-linearly.

Although Equation (5) might appear abstract, we show that it holds for a large class problems. Consider a Markov Decision Process (MDP) [35] $(\mathcal{Y}, \mathcal{A}, p, r, \eta)$, where $\mathcal{Y}$ is the state space, $\mathcal{A}$ is the action space, $p = \{p_h\}_{h \geq 1}$ a set of transitions kernels, $r = \{r_h\}_{h \geq 1}$ a set of reward functions, and $\eta \in [0, 1]$ is a discount factor. When an agent takes action $a$ in state $y$ at step $h$ it transitions to $y'$ with probability $p_h(y'|y, a)$, and receives a deterministic reward $r_h(y, a) \in [0, 1]$. Given a state $y \in \mathcal{Y}$, our goal is to find the action that maximizes the cumulative discounted reward over the next $H$ steps[4] Given an initial action, a way to compute a sample from its future cumulative discounted reward is to apply a Monte Carlo search algorithm (e.g., depth first search) truncating the planning depth at $\overline{h} < H$ to obtain a cheaper but biased estimate of the cumulative discounted reward. In this sense, this scenario fits the fixed-confidence MF-BAI setting. Let $M$ be the maximum depth, then, for each depth $m \in [M]$, we set as $\lambda_m$ the number of generated nodes, that, assuming a constant branch factor $\Lambda$, is exponential in $m$ (i.e., $\lambda_m = \Lambda^m$). Instead, for $\xi_m$ it is easy to obtain the following upper bounds on the maximum bias $\xi_m = (\eta^{m+1} - \eta^{M+1})/(1 - \eta)$. Now, it is possible to show that Assumption 1 holds (proof in Appendix C).

**Proposition 1.** *Consider $\Lambda \geq 2$. If $\lambda_m = \Lambda^m$ and $\xi_m = (\eta^{m+1} - \eta^{M+1})/(1 - \eta)$ for all $m \in [M]$, then Assumption 1 holds.*

## 4.2 Most Convenient Fidelity Thresholds

Given the structure of Theorem 1 and its interpretation, during a phase $m \in [M]$, we expect a good choice of thresholds $\boldsymbol{\alpha}$ to let IISE discard all the arms $i \neq 1$ that should (and can) be discarded at $m$. Therefore, during $m$, we aim at concluding that a given arm is suboptimal only if there is no other fidelity $\overline{m} > m$ in which gathering samples at $\overline{m}$ is more convenient, in term of identification cost, w.r.t. $m$. We formalize the previous concept in the following definition.

**Definition 1** (Fidelity Convenience). *Consider a sub-optimal arm $i \neq 1$ and a fidelity $m < M$ such that $\Delta_i > 4\xi_m$. We say that fidelity $m$ is convenient for arm $i$ if:*

$$\frac{\lambda_m}{(\Delta_i - 4\xi_m)^2} \leq \min_{\overline{m} > m} \frac{\lambda_{\overline{m}}}{(\Delta_i - 4\xi_{\overline{m}})^2}. \tag{6}$$

In Equation (6), we are comparing the cost of discarding $i$ gathering samples at phase $m$ only with the cost of discarding $i$ using a more precise fidelity $\overline{m} > m$. Therefore, solving it for $\Delta_i$, we identify the gaps of the arms that can be conveniently discarded at fidelity $m$ rather than at $\overline{m}$:

$$\Delta_i \geq \max_{M \geq \overline{m} > m} \frac{4(\xi_m \sqrt{\lambda_{\overline{m}}} - \xi_{\overline{m}} \sqrt{\lambda_m})}{\sqrt{\lambda_{\overline{m}}} - \sqrt{\lambda_m}} = \max_{M \geq \overline{m} > m} 4\xi_m + \frac{4(\xi_m - \xi_{\overline{m}})\sqrt{\lambda_m}}{\sqrt{\lambda_{\overline{m}}} - \sqrt{\lambda_m}}. \tag{7}$$

To discard all $i$ that satisfy Equation (7), it is sufficient to set thresholds $\alpha_m$ for $m < M$ as follows:

$$\alpha_m := \max_{M \geq \overline{m} > m} \frac{4(\xi_m - \xi_{\overline{m}})\sqrt{\lambda_m}}{\sqrt{\lambda_{\overline{m}}} - \sqrt{\lambda_m}}. \tag{8}$$

At this point, under Assumption 1, it is possible to prove that, with this peculiar threshold choice, the cost complexity of IISE can be rewritten in the following way (proof in Appendix C).

---

[3]More formally, $(\sqrt{\lambda_{i+1}} - \sqrt{\lambda_i})^2 \leq (\sqrt{\lambda_{j+1}} - \sqrt{\lambda_j})^2$ for $i, j \in [M - 1]$ and $j > i$.
[4]In the discounted setting, this is equivalent to search for an $\epsilon$-optimal action.

**Theorem 3.** *Under Assumption 1, selecting the thresholds $\alpha_m$ as in Equation (8), with probability at least $1 - \delta$, IISE returns the optimal arm with cost complexity $c(\tau)$ upper bounder by:*

$$c(\tau) \leq \widetilde{O}\left(\sum_{i=2}^{K} \min_{m \in [M]: \Delta_i > 4\xi_m} \frac{\lambda_m \sigma^2}{(\Delta_i - 4\xi_m)^2}\right) \leq \widetilde{O}\left(\sum_{i=2}^{K} \frac{\lambda_M \sigma^2}{\Delta_i^2}\right)$$

As we can appreciate, this result comes with two desired properties. First of all, IISE is provably robust w.r.t. algorithms that ignore the multi-fidelity structure of the problem. We remark that to have this nice property, Assumption 1 does not need to hold for each $m \in [M]$. It is always possible to pre-select a subset of fidelity for which the assumption holds. Secondly, the bound in Theorem 3 closely mimics the behavior of the lower bound (Equation 2). The only difference stands in the fact that $2\xi_m$ is now replaced with $4\xi_m$. We leave this gap to be filled in for future work. Notice, again, that the similarity also holds for the case in which we select a subset of fidelity for which the assumption holds. In this case, however, our minimization will be on a restricted fidelity set w.r.t. the original one.

## 4.3 Nearly-Optimal Identification

As for the Successive Elimination algorithm, one can easily modify IISE so that it returns an $\epsilon$-optimal arm. In this case, we can show that the identification cost is upper bounded by (proof in Appendix C):

$$c(\tau) \leq \widetilde{O}\left(\sum_{i \notin \mathcal{K}_\epsilon} \min_{m: \Delta_i > 4\xi_m} \frac{\lambda_m \sigma^2}{(\Delta_i - 4\xi_m)^2} + \sum_{i \in \mathcal{K}_\epsilon} \min_{m: \epsilon > 4\xi_m} \frac{\lambda_m \sigma^2}{(\epsilon - 4\xi_m)^2}\right),$$

where $\mathcal{K}_\epsilon := \{i \in [K] : \mu_i > \mu_1 - \epsilon\}$. As usual, for arms that are not $\epsilon$-optimal we pay the original cost of Theorem 3. For the ones in $\mathcal{K}_\epsilon$, instead, their cost is actually clipped by $\epsilon$. In multi-fidelity, this has the further implication that, depending on the values of biases, costs and required accuracy, we might never query the most expensive unbiased fidelity $M$. In particular, let us focus on a two-armed bandit problem with two fidelity values, and consider $\Delta_2 < \epsilon$. Then, we will never query $M$ if:

$$\epsilon \geq 4\xi_m + \frac{4\xi_m \sqrt{\lambda_m}}{\sqrt{\lambda_M} - \sqrt{\lambda_m}}.$$

The first term guarantees that IISE can discard $i$ using $m$, the second one, instead encodes the convenience of $m$ w.r.t. to $M$ in terms of costs. As $\lambda_M$ increases, this term shrinks to zero. When $\lambda_M \to +\infty$, the condition reduces to $\epsilon > 4\xi_m$. This means that if fidelity $m$ is precise enough for the desired accuracy $\epsilon$, we completely neglect fidelity $M$.

## 5 Breaking the Cost Barrier

We now show that (a slightly modified version of) IISE can further reduce the cost complexity when more fine-grained knowledge on the type of error that approximators introduce is available. The main intuition behind the following reasoning is that what matters when exploiting fidelity $m < M$ is not the bias they introduce, but whether the arms do not preserve the order w.r.t. the mean of the true reward distributions $\nu_{i,M}$. Indeed suppose to have access to a fidelity with a very large bias but which fully preserves the order on the means of the arms. In this scenario, it is possible to run a standard BAI technique on this fidelity *only* and still obtain the optimal arm. We now formalize how much a certain fidelity $m$ preserves the order with the novel concept of *maximum bias variation* $\gamma_m$:

$$\gamma_m := \max_{i,j \in [K]} \{|\xi_{i,m} - \xi_{j,m}|\}.$$

Under the assumption that $\gamma_1 \geq \gamma_2 \geq \cdots \geq \gamma_M := 0$ are available to the learner, IISE can be conveniently modified to obtain the following bound on the cost complexity (proofs, pseudocode, and

formal statement in Appendix D):[5]

$$c(\tau) \leq \widetilde{O}\left(\sum_{i=2}^{K} \min_{m \in [M]: \Delta_i > 2\gamma_m} \frac{\lambda_m \sigma^2}{(\Delta_i - 2\gamma_m)^2}\right).$$ (9)

To obtain such a result, the only required modifications stand in the elimination condition, which now becomes $\hat{\mu}_{j,m,t} - B\left(t, \frac{\delta}{KM}\right) \geq \hat{\mu}_{i,m,t} + B\left(t, \frac{\delta}{KM}\right) + \gamma_m$, and in the definition of the thresholds $\alpha_m$, in which $4\xi$'s are swapped with $2\gamma$'s. We will refer to this modified algorithm with IISE-$\gamma$. As our proofs show, IISE-$\gamma$ can also exploit upper bounds $\phi_m$ on the true maximum bias variation (i.e., $\phi_m \geq \gamma_m$); the complexity is as in Equation (9), but $\gamma_m$ is replaced with $\phi_m$.

We remark that, by definition, $\gamma_m \leq 2\xi_m$, which provably demonstrates the benefits of exploiting order-aware knowledge in the MF-BAI setting. Interestingly, we also notice that in the worst-case scenario (i.e., $\gamma_m = 2\xi_m$), Equation (9) implicitly recovers the cost complexity of Theorem 3.

**Practical Relevance of Maximum Bias Variation.** Finally, we highlight that the knowledge about upper bounds $\phi_m$ is available in a large class of problems. In particular, consider settings in which means of fidelity $m < M$ are underestimations of the ones of fidelity $M$, i.e., $\mu_{i,M} - \mu_{i,m} \geq 0$ for all $i \in [K]$ and $m \in [M]$. This is the case, for instance, of the stochastic planning application of Section 4, but it also common in other settings such as training models with less iterations. In these problems, it always holds that $\gamma_m \leq \xi_m$ (proof in Appendix D). Therefore, we can set $\phi_m = \xi_m$ in IISE-$\gamma$ to obtain the improved cost complexity of Equation (9), where $2\gamma_m$ is replaced with $2\xi_m$.

## 6 Related Works

In this section, we revise the literature with particular attention to best-arm identification and the multi-fidelity setting.

**Best-Arm Identification.** The BAI setting has aroused the interest of the research community for a long time [10, 6, 2]. In particular, [10] proposed the Successive Elimination algorithm, together with the first gap-dependent upper bound on the sample complexity: $O(\sum_{i=2}^{K} \Delta_i^{-2}(\log(\delta^{-1}) + \log(K) + \log(\Delta_i^{-2}))$. This algorithm is known to match the lower bound up to logarithmic factors. Indeed, [29, 24] showed that, for all possible instances, every $\delta$-correct algorithm requires at least $\Omega(\sum_{i=2}^{K} \Delta_i^{-2} \log(\delta^{-1}))$ samples to output the optimal arm with probability $1 - \delta$. To overcome the logarithmic mismatch, several works have proposed both tighter upper [12, 23, 16, 7, 13, 8] and lower bounds [16, 13, 8]. Concerning the BAI literature, we leverage on the change of distributions arguments presented in [24] to build lower bounds for our novel cost complexity metric. As for the upper bound, instead, to favor the simplicity of exposition and intuition, we build on the Successive Elimination algorithm. Clearly, our proofs can straightforwardly incorporate the techniques from the more refined analysis [e.g., 23] to tighten the logarithmic dependencies.

**Multiple Fidelity.** The use of multiple fidelity has gathered particular attention from the Bayesian Optimization (BO) [5] field, with a large variety of algorithms that operate under different assumptions and with different goals [15, 33, 34, 28]. In particular, a recent line of works [22, 19] has focused on minimizing the *simple regret* suffered when a certain budget has been spent to optimize a target function under smoothness conditions guaranteed by Gaussian processes [37]. The study of the simple regret has also been extended to black-box function optimization [39, 40, 11] under hierarchical partition assumptions on the target function. However, all previous work differs significantly from the study we present in this paper. Indeed, they optimize different goals w.r.t. our novel *cost complexity* notion and make structured assumptions about the target function (i.e., the arm space). With most of them [21, 20, 39, 11], we share the assumption on the knowledge of the upper bound on the maximum bias $\xi_m$ for each fidelity. Of particular interest w.r.t. our work, is the one of [11]. In particular, the authors highlighted the concept that what matters when exploiting fidelity $m < M$ is how much they preserve the order rather than the bias they have. However, their assumption about order preservation is expressed in a general form; with the maximum bias variation $\gamma_m$, instead, we

---

[5]The cost complexity bound is obtained under an assumption equivalent to Assumption 1 in which all $\xi$'s are replaced with $\gamma$'s.

Table 1: Cost complexity results (mean and $95\%$ confidence interval of 100 runs).

| Algorithm | Synthetic A | Synthetic B | Yahtzee |
|---|---|---|---|
| MFE | $212.43 \pm 3.89$ | $3330.27 \pm 62.59$ | $342.75 \pm 0.69$ |
| IISE | $11.47 \pm 0.31$ | $27.06 \pm 0.77$ | $0.53 \pm 0.01$ |
| IISE-$\gamma$ | $2.53 \pm 0.06$ | $6.39 \pm 0.17$ | $0.001 \pm 0.01$ |

propose a closed-form expression for this concept, and we quantitatively show its impact in terms of benefits with respect to the upper bound on the maximum bias. We also showed that upper bounds $\phi_m$ on this metric are known and available in a large class of problems, and that our algorithm can successfully exploit this knowledge to reduce identification cost.

The work most closely related to ours is [21]. The authors consider the finite arm setting with known upper bounds on the maximum bias for each fidelity. However, while we take into account the identification cost, in [21], the authors analyze a *pseudo-regret* notion that sets as an optimal strategy the one that invests the whole budget in querying the optimal arm. Interestingly, we got similar results, in terms of structure, for what concerns the lower bound, although we remark that the proof techniques are different since we deal with a different performance metric. As for the proposed algorithm and the upper bounds, our proofs rely on novel concepts such as the fidelity convenience (i.e., Definition 1), which are tailored to the identification cost we have analyzed. In addition, as we have seen in Section 4, we make a different assumption w.r.t. [21]: our assumption directly states a relationship between costs and biases, which is more reasonable since the goal of the multi-fidelity setting is to trade-off between the two. Finally, w.r.t. [21], we analyze the effect of more fine-grained knowledge on different fidelity, which provably leads to performance improvements.

To conclude, some works [9, 30, 41] have also investigated the use of multi-fidelity simulators in Reinforcement Learning (RL) [42] domains. Among these works, [9] develops fixed confidence algorithms (up to a desired precision $\epsilon > 0$) both for the RL setting and for the bandit one. However, their setting is significantly different from ours as they do not consider each fidelity to be related to a given cost. Instead, they simply aim at minimizing the number of sub-optimal steps at fidelity $M$, while controlling the number of samples taken in fidelity $m < M$ to ensure they are polynomial.

## 7 Numerical Validation

In this section, we provide a numerical validation of our theoretical claims in both synthetic and simulated domains. To prove that IISE is successful at exploiting fidelity, we compare it with: (i) Successive Elimination (SE) [10], which exploits only maximum fidelity, and with a version of IISE that chooses the thresholds $\alpha_m$ so as to exploit every fidelity as much as possible (i.e., $\alpha_m$ close to 0). We call this modified version Maximum Fidelity Exploit (MFE) algorithm. Table 1 reports all the results in terms of the percentage of cost complexity of a given algorithm w.r.t. the one of Successive Elimination. The value 100 corresponds to the relative cost complexity of SE.

**Synthetic bandits.** We start presenting some experiments on two synthetic bandits (Synthetic A and Synthetic B) with randomly generated arms. These experiments aim at validating our robustness results along with the benefit of the order-aware knowledge. Synthetic A setting parameters are $K = 2000$, $M = 4$, $\lambda = [1, 10, 100, 1000]$, $\xi = [1.15, 0.225, 0.015, 0]$, $\gamma = [0.3, 0.05, 0.001, 0]$; for Synthetic B, instead, $K = 1000$, $M = 5$, $\lambda = [16, 64, 256, 1024, 4096]$, $\xi = [1.15, 0.45, 0.105, 0.0105, 0]$, $\gamma = [0.3, 0.1, 0.01, 0.001, 0]$. To make the $\gamma$ and $\xi$ settings directly comparable, we generated the arms such that the fidelity index $m$ is left unchanged (i.e., it simultaneously holds that $\lambda_1 > \lambda_2 > \cdots > \lambda_M$, $\xi_1 > \xi_2 > \cdots > \xi_M$, and $\gamma_1 > \gamma_2 > \cdots > \gamma_M$). Further details on this generation process can be found in Appendix E. As we can appreciate from Table 1, IISE substantially reduces the cost complexity w.r.t. SE, and, its order-aware version IISE-$\gamma$, obtains further improvements. Table 1 also highlights the importance of choosing a good threshold $\alpha_m$. Indeed, MFE insists in exploiting a given fidelity, even when there is no convenience in doing so, thus obtaining significantly worse results than SE.

**Yahtzee.** We now apply our algorithm to the problem of choosing the first action in the stochastic planning domain introduced in Section 4. More specifically, we consider the Yahtzee game [3]. Yahtzee is a sequential dice game in which a player rolls 5 dice up to 3 times and then chooses a particular move that assigns a score to the final dice combination. For instance, typical moves are:

"Sixes" with a score given by the sum of dice with the number 6, and "Yahtzee", that assigns 50 points if all the dice show the same number, and 0 otherwise. The total number of these possible moves is 14 (i.e., our branching factor $\Lambda$). The game proceeds in 13 rounds. During each round $t$, the player selects a move and obtains the corresponding points. We consider the variant of Yathzee in which, if the player has already selected a certain move at $\bar{t} < t$, then, at $t$, that move will lead to reward 0. Furthermore, during each round, the player will roll the 5 dice only once, after which a move has to be selected. Further details are provided in Appendix E. Table 1 shows that our method significantly reduces the total number of generated nodes, thus lowering successfully the total cost of identifying the best action, and, that exploiting the upper bounds on the maximum bias variation instead of $\xi$'s is beneficial in terms of total cost. Finally, MFE overexploits fidelity and underperforms SE.

## 8 Conclusions and Discussion

In this work, we have studied, for the first time, the variant of the fixed-confidence BAI scenario in which the agent has access to multiple approximations with different costs and biases of the original bandit model. We proposed the novel notion of cost complexity to evaluate the performance that a given algorithm obtains in these situations. We derived a lower bound on this index, which shows several interesting properties of the problem, along with its main challenges. We also presented a novel algorithm, IISE, which provably reduces the cost complexity compared to algorithms that ignore the availability of multiple fidelity. More specifically, IISE discards a sub-optimal arm $i$ with the most convenient, in terms of cost and fidelity, thus closely mimicking the shape of the lower bound. Furthermore, we also showed that the maximum bias is not what actually matters in identifying the best arm. Indeed, better guarantees can be obtained with the knowledge of the maximum bias variation, an original concept that we introduce to encode how much a given fidelity preserves order.

To close the problem from a theoretical perspective, it is reasonable to assume that both upper and lower bounds could be improved in settings that do not fall under Assumption 1. More specifically, the lower bound of Theorem 1, that considers the minimum over the entire fidelity set, seems currently optimistic since it does not encode the cost that is needed to explore at other fidelities. To tackle this issue, proof techniques that rely on the characteristic time function [13] might lead to tighter lower bounds. Consequently, adapting to the MF setting their well-known "Track and Stop" strategy [13] might provide upper bounds that asymptotically match the lower bound in its characteristic time form. However, results obtainable from these approaches usually come at the cost of lower interpretability (and, depending on the nature of the resulting characteristic function, computational complexity). On the other hand, Theorem 1 provides several intuitions behind the challenges of the Multi-Fidelity BAI setting, and IISE is computationally efficient.

Furthermore, whenever Assumption 1 does not hold, the cost of "exploring" lower fidelities becomes more and more relevant. Indeed, Assumption 1 guarantees that there is a large enough "fidelity gap" (i.e., fidelity differs in terms of the joint costs-biases relationship specified by the assumption). The main issue in settings, where this gap is small, roots down to the chicken-egg problem of MF-BAI. On one side, we would like to exploit these approximators to discard sub-optimal arms; on the other side, we do not know if these approximators will be useful (i.e., in the worst case, we pay an exploration cost). In the case, where the "fidelity-gap" is small, this problem is repeated over and over for all these fidelities close to one another (i.e., the cumulative sum of the exploration costs becomes non-negligible). In this sense, tighter lower bounds (such as the ones previously suggested) might shed light on whether such assumptions are necessary or not, or if tighter assumptions are possible/necessary. This represents an exciting line for future research.

To conclude, we note that previous studies [45, 18, 1] highlighted the strong connections between fixed-confidence BAI and learning near-optimal RL policies while minimizing the number of samples gathered in the environment. In this sense, our ideas pave the way for future work that aims at learning these policies when multiple simulators are available. Indeed, previous efforts in MF-RL [9, 30, 41] do not directly take the challenging cost-precision trade-off into account.

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
