# OpenReview forum: "Multi-Fidelity Best-Arm Identification"
_NeurIPS.cc/2022/Conference — NeurIPS 2022 Accept_

### Official Review · Reviewer_LAgF · 2022-06-22

**Rating:** 3
**Confidence:** 3
**Soundness:** 3 good
**Presentation:** 1 poor
**Contribution:** 2 fair

**Summary:**

The authors tackle establishing the best arm in a multi-arm bandit problem, where each arm can be pulled with a different “fidelity”.  Lower fidelity evaluations are cheaper to evaluate, but provide a more biased estimate of the objective function.  They present and analyse a new algorithm, IISE, that theoretically outperforms basic bandit algorithms that do no exploit the multi fidelity structure of the problem.  They then numerically confirm this on two experiments.


**Questions:**

I raised several questions in my main review.


**Limitations:**

There are no ethical considerations.

I raised several comments on the limitations of the work above, including the limitations of the setting, the approach taken, the presentation and the empirical validation.


**Strengths And Weaknesses:**

====== I posted this in a comment after my review, but it isn't visible to the authors for some reason.  Sorry for the confusion.

## Slight revision.

I have thought about this paper and my review a little more since initially writing it.  I figure I'd say this in the discussion phase anyway so I might as well put it here so the authors can respond in their rebuttal.

My stance on the paper has relaxed a little bit.  While I still broadly stand by my concerns, I think if the authors can allay them, then I would endorse this paper.

Specifically:
- I believe the paper could benefit from simplification of the exposition, maybe even cutting some material to the supplement.  I would consider adding an executive summary of the method, banner figure, algorithm/pseudo-code block*, flow chart etc.  Right now it is an impenetrable wall of math.
- The fixed subgaussian noise ($\sigma^2$) still troubles me.  I think this is my main technical reservation.
- The empirical validation is very weak.  Right now the paper is hinging on the significance of the theoretical and methodological contribution, as it hasn't, in my opinion, been shown empirically _beyond doubt_ to be better than the alternatives.

Thanks, LAgF

*I know you have an algorithm block in there, but it is still quite low-level.  I'd like to see a higher-level / word-based / pseudocode version of this.



========= Original Review:


I will preface my review with the statement that I am not an expert on bandits, or on multi fidelity methods.  That said, the paper is well written enough on a high level that I get the general point.

There is a huge amount of theoretical work that has gone into this paper, and for that, the authors are to be commended.  I think there is definitely an audience at NeurIPS that will get a lot from reading this paper and the accompanying materials (I only skimmed the results presented in the supplement).

That said, I am not currently endorsing the publication of this paper, as I have some fairly substantial questions.  If the authors can allay my concerns and make some (again, fairly substantial) updates to the paper, then I would be willing to endorse it.

I query some of the modelling choices made (1).  I also struggle with some elements of the presentation (2), and the empirical validation (3) .  I will try and break my criticism down a bit.

# Major Comments:

### 1.  Fixed subgaussian noise.
You assume a fixed variance for the subgaussian noise across all arms.  This seems like a pretty fundamental limitation to me.  I often think about mutlifidelity in terms of how many iterations / MC samples I allow my simulator/solver to take.  Therefore, lower fidelity simulators have a much higher variance on their evaluation than higher fidelity simulators.  Alternatively, lower-fidelity simulators may have lower variance because they are too simple to capture the full variability.  Finally, in the MDP example you give, longer windows have a higher variance, and therefore this doesn’t apply.  As the variable evaluation noise operating regime is not catered for in your work, it therefore excludes a large set of practical application domains where your theoretical guarantees don’t hold.  I invite the authors to comment on this.


### 2.  I really struggle with the presentation of the material.
I understand that bandits and MF methods are individually quite notationally dense and difficult to get down on paper; and that a bandit + MF method is probably that difficulty squared.  However, the authors must do better (in my opinion) to strip out all the complexity and present the ideas as simply and as notationally clear as possible.

- This paper may benefit greatly from a banner diagram to try and visually present the ideas, and at least unload some of the burden of explaining everything from the text.

- For instance, the aside on MDPs and Prop 1 doesn’t seem to be used.  This should be cut from the main text and placed entirely in the supplement to clear space and reduce the amount of technical content.

- A table of notation in the supplement would be very helpful as well.

- Really try to keep subscripts, superscripts, diacritics to a minimum.


### 3.  Weak empirical validation
There is obviously a sizeable theoretical contribution in this paper, however, the empirical validation is lacking in my opinion.  A toy synthetic problem and a single other problem tested against a single baseline doesn’t really convince me that your method works and is superior.  Obviously this isn’t required for publication, but, I do not consider this method to have been empirically validated.  To convince me of this, I would want to see a few more baselines evaluated on a range of more standard problems on which existing methods have been tested.  I have little-to-no idea if “X” is a good score on “synthetic toy problem 2”, whether synthetic toy problem 2 is a fair baseline on which to apply MFE etc.  I imagine there is a wealth of domains you can leverage to get a good example – image classification with different length glimpse sequences [Harvey, Teng, Wood, 2019], maybe train different stochastic classifiers with different sizes, number of latent samples, numbers of particles etc.  Maybe we just actually use the MDP / RL examples?  I really think this paper would be strengthened by using a domain that many readers will “understand.”  There is also a chronic shortage of implementation and experimental details in the main text and the supplement.  I do note however that code was released.


# Minor Comments.
None, actually.


# Very Minor / Typographical Comments.
i. Theorem 3:  “upper bounder” -> “upper bounded.

ii.  Throughout:  (I.e. in Theorem 3)  Inconsistent use of “Equation \\eqref\{\}”, “\\eqref\{\}”, “Equation \\ref\{\}” and just “\\ref\{\}”.

iii.   Line 232:  “w.r.t” -> “w.r.t.”.

iv.  Line 243: “0” -> “zero”  (numbers less than 13 are typically written out).

v.  Try to avoid footnotes.  They are bad for readability.  If they are important then they should be integrated into the main text, otherwise, they should be removed.

vi.  Line 454:  Matyas shouldn’t be “et al.”.  Also inconsistent use of just first initial.


# Summary.
As outlined above, I think this paper has some serious limitations and weaknesses as it is currently presented.  I think there is a good conference paper in there somewhere, but I do not believe it is currently at that point.  There could even be a better, more impactful, and more widely distributed conference paper if published at a more specialised venue, if you are unfortunately rejected in this review cycle.  Good luck.

---

> ### Author Response · Authors · 2022-08-02
> **Theoretical empirical validation + review summary**
>
> **Empirical validation**. First of all, we would like to highlight the following crucial aspect. The goal of our experiments is not to prove that our method is the best possible one in terms of empirical performance across a broad number of domains and baselines. On the other hand, as usually done in theory-oriented works, we focus on simple domains with the goal of \emph{empirically verifying the theoretical claims}. In particular, our experiments focused on:
>
> * Showing the robustness w.r.t. Successive Elimination.
>
> * Showing that setting the thresholds in a theoretical principled way (as IISE does) helps w.r.t. trying to exploit a fidelity as much as possible (as MFE does). Setting the thresholds is the crucial aspect of our approach, since the choice directly trade-off between fidelity exploration and exploitation. Trying to discard each possible element with a given fidelity $m < M$ clearly reduces the number of calls to the precise bandit model $\nu_M$, but, as our experiments show, it can introduce a significant exploration overhead in the cost complexity.
>
> * Showing that order-aware knowledge (i.e., using $\gamma$ or $\phi$) reduces the identification cost.
>
>
> In this sense, our experiments are similar to the ones that are usually done in theoretical works. See, for instance, "The Multi-Fidelity Multi-armed Bandit" (Kandasamy et al., 2016).
> That being said, we now make a couple of additional comments:
>
> * *On the domains.* Synthetic domains are usually adopted in empirically validation of theoretical bandit algorithms. These sort of domains have also been employed in the work that is the most related to ours (i.e., "The Multi-Fidelity Multi-armed Bandit" Kandasamy et al., 2016).
>
> * *On additional MF baselines.* Concerning the baselines, first of all, one might try to insert baselines from the MF literature. We notice, however, that our work is the only one that performs best-arm identification in multi-fidelity domains. Applying, for instance, MF-UCB ("The Multi-Fidelity Multi-armed Bandit" Kandasamy et al., 2016) to our setting requires appropriate modifications (e.g., an appropriate elimination rule to discard the arms) together with theoretical analysis proving that these modification are sound (i.e., the optimal arm is identified w.h.p.). All the other MF works, moreover, make structural assumptions on the arm-space that make them not applicable in our setting. Furthermore, as MF-UCB, they would also suffer the previous problem: how to turn them into BAI algorithms if they have not been designed for such purpose? Which stopping condition should they employ? Notice that the stopping condition has clearly a significant impact on the performance (and the soundness) of the algorithm.
>
> * *On additional BAI baselines.* Secondly, one might want to add BAI baselines. We remark, in this sense, that our algorithm builds its theory upon the one of Successive Elimination, which is a well-known best-arm identification approach. Our proofs, however, could also have been extended using best-arm identification methods that make uses of more sophisticated analysis, which might come together with better empirical performance. However, extending many literature BAI algorithm to the MF setting to empirically compare their performance clearly falls outside the scope of our work. Comparing the performance of our MF method with a BAI algorithm that uses a more involved theory would be unfair since its potential benefit arises from more involved theoretical tools/design, rather than being better at exploiting the MF structure of the problem.
>
> * *Implementation details.* Given the "software-engineering simplicity" of best-arm identification algorithms in unstructured arm-space domains, the implementation follows directly from the pseudo-code. To speed up the experiment execution time, we have made our code parallizable so that it is possible to run multiple runs of the same domain over multiple cores. Concerning the hyper-parameters, we have spotted that the confidence level $\delta$ was missing from the appendix. We have now included it in the new version that we uploaded.
>
>
> **On the summary of the review.** To conclude, we would like to remark that, as the reviewer also acknowledges at the beginning of its review, "there is definitely an audience at NeurIPS that will get a lot from reading this paper and the accompanying material". Moreover, we stress that previous theoretical works on multi-fidelity bandits have been published at NeurIPS (e.g., "The Multi-Fidelity Multi-armed bandit", Kandasamy et al., NeurIPS 2016). For these reasons, we believe that this venue is appropriate for our work.
>
> Finally, we thank the reviewer for typographical comments and typos. We have fixed them in the version we have uploaded. We hope that our comments have shed lights on the doubts exposed by the reviewer. We will be glad to provide additional clarifications if the reviewer has further questions/doubts/requests.

---

> ### Author Response · Authors · 2022-08-02
> **Presentation of the material**
>
> **Presentation of the material**
> Concerning the presentation of the materials, we acknowledge that the notation might not be easy to follow for non-experts in bandits and multi-fidelity methods. However, we would like to remark that our notation is consistent with the one adopted in the literature (see, for instance, "The Multi-Fidelity Multi-armed Bandit" Kandasamy et al., 2016).
> However, as suggested by the reviewer, we believe that a table in the supplementary material might benefit the reader. We added it in the recently uploaded version. Moreover, in Appendix F, the reviewer can now find a banner diagram of the proposed algorithm, together with a simplified (and we hope, intuitive) explanation of the approach. We will modify the main text to include the diagram together with the description for the final version of the paper.
>
> Concerning the discussion on the MDP and Proposition 1, we believe its contribution to be relevant and its presence necessary in the main text. Indeed:
>
> * It shows a relevant and general application in which Assumption 1 holds.
>
> * A planning application is studied in our experimental section. Removing the discussion from Section 4, will require a) the description of the MDP planning example in the experiments, b) a discussion on why/how we can apply our method to this setting , i.e., proving that Assumption 1 holds  in the experimental setting. Given, however, the generality of the application, we believe that it better fits in Section 4, providing additional theoretical contributions.

---

> ### Author Response · Authors · 2022-08-02
> **Subgaussian random variables**
>
> We thank the reviewer for the effort to review our paper. Please, find below our answers and comments to the raised doubts. We hope they will help in clarifying some crucial aspects of our work.
>
> **Fixed sub-Gaussian assumption.**
> First of all, we would like to remark that the fixed sub-Gaussian assumption is the one that has been dealt with in previous MF studies such as "The Multi-Fidelity Multi-armed Bandit" (Kandasamy et al., 2016). With such an assumption, the main goal of the problem stands in understanding the *opportunities and challenges that arise when dealing with **biased** approximators*. Understanding the effect that this bias introduces has also been at the heart of several other works in the MF literature (e.g., "Gaussian process bandit optimisation with multi-fidelity evaluations", Kandasamy et al., 2016, "Multi-fidelity black-box optimization with hierarchical partitions", Sen et al., 2018, "Adaptive multi-fidelity optimization with fast learning rates", Fiegel et al., 2020). In all these works, the authors focus solely on understanding the biases that the approximators have (they either assume fixed noise or even deterministic functions) by assuming access to upper-bounds on the maximum bias $\xi$ that each fidelity introduces. In this sense, our setting is **well-established** in the literature. The goal lies in understanding how to **exploit biased approximators**.
>
> As our work and the literature shows, dealing with biased approximators requires facing multiple non-trivial challenges. We believe that the setting in which $\sigma$ is fidelity-dependent (i.e., $\sigma_m$) lies outside the scope of the present paper.
>
>
> For what concern the MDP/planning example, instead, we highlight the following facts:
>
> * Its main purpose is to show a concrete application example in which the biases and costs have shapes that satisfy Assumption 1.
>
> * We can still safely apply our method using, for each fidelity, the pseudo-variance of the deepest horizon (which is what we have done in our experiments). This still ensures the theoretical properties of our analysis. In particular, we notice that the robustness property of Theorem 3 holds since a general algorithm that ignores the multi-fidelity structure of the problem requires the pseudo-variance of the deepest horizon (which, indeed, is the largest one).

---

> ### Author Response · Authors · 2022-08-06
> **Follow up**
>
> We wanted to follow up on the response to the previous comments. In particular, are the reviewer’s doubts concerning subgaussian r.v. / presentation of the materials / empirical validation of theoretical claims solved? If yes to the above, is the reviewer satisfied with the overall response? If no, would the reviewer be willing to engage in further discussion about the disagreements? Thanks again for your effort and comments!

---

> ### Comment · Reviewer_LAgF · 2022-08-07
> **Rebuttal Response**
>
> To the authors,  Sorry for the delay in my response.  Thank you for your response to my review, to the other reviews, and for engaging in the discussion.
>
> 1) Presentation:  The authors have added a figure in the supplement that goes some of the way to explaining the method.  It could be made smaller for the final version, but it will definitely help tell the story.  There is also a table of notation.  Minor edits were made to the main text, but they are mainly fixing mistakes, as opposed to working on the clarity that I felt was missing.  I still feel like there is work to be done simplifying and stripping down the paper to highlight the core of the contribution.  I do consider my concerns on presentation to have been downgraded to a 'minor concern'.
>
> 2) Fixed subgaussian noise:  I understand the authors comments on the fixed subgaussian noise.  Understanding how to use biased approximations is important.  However, I do not feel that "variable uncertainty has been studied elsewhere" is a valid justification for considering a fixed variance.  Especially when multi-fidelity simulators, very clearly, do not have a fixed variance.  I understand there may be work-arounds, such as assuming the maximum variance.  Establishing the variance of an estimator is also non-trivial, especially for complex state-spaces and dependency structures.  I still consider the fixed subgaussian noise to be a major, and near-terminal, weakness of the work.
>
> 3) Empirical validation:  I again understand that the authors are trying to show that SE abides the theory.  However, I do not believe this is necessarily what you have shown.  You have shown that SE outperforms MFE on three (really two) tasks.  To show that the theory holds, I would look to show things like, for instance, that the bound holds in practice (probably on a toy problem that is analytically tractable), how tight is the bound as a function of the hyperparameters / model, are the known degenerate scenarios where the estimator breaks down, does it break down as the theory predicts etc?  As a result, I do not consider your method to have been thoroughly empirically validated as a purely methodological contribution, nor, your theory rigorously proven out with experiments as you assert.  Therefore, it is purely a theoretical contribution under some relatively strict constraints.  My concerns about empirical validation remain.
>
> As such, I am (at least for now) going to stay with my original scoring.  I implore the authors to seek out challenging and interpretable environments in which to test their method.
>
> Thank you and good luck.
> LAgF.

---

> > ### Author Response · Authors · 2022-08-08
> > **Empirical validation**
> >
> >
> > We frankly disagree with the reviewer on this point. Given the theory, our experiments provide empirical evidence of the three claims that we listed in our previous comment.
> >
> > Furthermore, in our opinion, showing that the "bound holds in practice" is not interesting since there are proofs that demonstrate the validity of the bound.
> >
> > The sentence that concerns visualizing "the bound as function of the hyperparameters / model, are the known degenerate scenarios where the estimator breaks down, does it break down as the theory predicts etc?" is unclear to us. More specifically, there are no relevant hyper-parameters in our algorithm. The only term that, in some sense, can be considered a hyper-parameter is $\delta$: a) its dependency is well-known to be logarithmic in finite-arm BAI scenarios (both in uppers and lower bounds); b) in our experiments, we have already set it close to $0$, which is the hardest scenario (in the experiments, indeed, all algorithms identified correctly the optimal arm in each of the $100$ runs). Moreover, there is no "model" in our algorithm, but just a matrix of estimators. Finally, it is unclear to us what the reviewer means by "degenerate scenarios in which estimators breaks down".
> >
> > If the reviewer meant to plot the upper bound as a function of $\xi$ and $\lambda$, we believe that it provides no additional contribution to the formula, which is already clearly stated within the theorems. If the reviewer believe it is necessary, we can add such plots into the appendix. The same argument holds for what concerns showing that the bound holds in practice. However, as explained above, we do not believe that such experiments will provide additional contributions. Furthermore, if the reviewer is willing to provide a well-defined precise experiment of such a theoretical nature, we would be happy to deliver the results of such an experiment.
> >
> > Finally, concerning the "challenging and interpretable environments", we would like to remark that the Yathzee domain (and the planning application itself) takes a step further w.r.t. previous empirical validation in finite-arm multi-fidelity bandits.
> > Indeed, it is significantly more complex than standard empirical settings employed in previous papers (e.g., "The Multi-Fidelity Multi-Armed Bandit, Kandasamy et al., 2016).
> > As shown in "Reinforcement learning benchmarks and bake-offs II" (NIPS 2005), the size of state-space representations of the MDP associated with the Yathzee problem are in the order of hundreds of millions (and with $14$ actions), which is remarkably large for a tabular MDP.

---

> > ### Author Response · Authors · 2022-08-08
> > **On subgaussian random variables**
> >
> > We invite the reviewer to ponder on the claim: "fixed-subgaussian noise assumption is a major, and near-terminal, weakness of the work". First of all, we highlight that such criticism is not a criticism to our work but a criticism to an **entire strand of well-established literature**, that has been **published at top-tier venues** in recent years. For instance:
> >
> > * "Gaussian Process Bandit Optimisation with Multi-fidelity Evaluations", Kandasamy et al., NIPS 2016. Fixed noise setting.
> > * "Multi-fidelity multi-armed bandit" Kandasamy, NIPS 2016. Fixed noise setting.
> > * "Multi-fidelity Bayesian Optimisation with Continuous Approximations", Kandasamy et al., ICML 2017. Fixed noise setting.
> > * "Multi-fidelity Gaussian Process Bandit Optimisation", Kandasamy et al., JAIR 2019. Fixed noise setting.
> > * "Multi-Fidelity Black-Box Optimization with Hierarchical Partitions", Sen et al., ICML 2018. Deterministic setting.
> > * "Noisy Blackbox Optimization using Multi-fidelity Queries: A Tree Search Approach", Sen et al., AISTATS 2019. Fixed noise setting.
> > * "Adaptive multi-fidelity optimization with fast learning rates", Fiegel et al., AISTATS 2020 . Deterministic setting.
> > * "A General Framework for Multi-fidelity Bayesian Optimization with Gaussian Processes", Song et al., AISTATS 2019. Fixed noise setting.
> >
> > All these works have clearly provided significant contributions to the multi-fidelity field, **focusing their analysis on the bias while assuming fixed subgaussian noise (or even deterministic settings)**.
> >
> >
> > To dive deeper into the discussion, one of the reasons of studying this setting with fixed subgaussian noise is that extensions to problems in which $\sigma$ is fidelity-dependent are direct. As also discussed in some of the previous works (see for instance, "Multi-fidelity Gaussian Process Bandit Optimisation", Kandasamy et al., JAIR 2019), studying biased approximators provides several theoretical tools and insights into how to develop methods for cases in which the noise is fidelity-dependent. Our algorithm and analysis requires minimum modifications to be extended to the case in which $\sigma$ is replaced with $\sigma_m$. The proof of the lower bound is mainly left unchanged (just replace $\sigma$ with $\sigma_m$). For the upper bound, one could adopt the appropriate Hoeffding inequality, and, consequently, the equivalent concept of fidelity convenience. This sort of modifications are straightforward. The formulation that we provided, on the other hand, significantly simplifies the exposition, which already uses, by construction, a rather dense notation. If the reviewer believes this is necessary, we will insert a note in the appendix to clarify this concept.

---

> > ### Author Response · Authors · 2022-08-08
> > **Presentation and algorithm outline**
> >
> > We thank the reviewer for having appreciated our effort in making the paper more accessible to the reader. Additionally, we highlight that an algorithm outline (as previously requested) was added to simplify the exposition in Appendix F. Since the reviewer did not mention it in the response, we remark it in case it was missed while reading the new version of the supplementary materials.

---

### Official Review · Reviewer_K1jV · 2022-07-06

**Rating:** 7
**Confidence:** 3
**Soundness:** 3 good
**Presentation:** 3 good
**Contribution:** 3 good

**Summary:**

This work studies the multi-fidelity variant of the best-arm-identification problem. It first proposes a new performance measure called cost complexity, then gives a lower bound for the cost complexity of this problem. To solve this problem, this work offers an algorithm called Iterative Imprecise Successive Elimination (IISE), along with the cost complexity upper bound analysis. Then this work gives a better result when the algorithm has access to some more fine-grained knowledge of the multi-fidelity setting. Finally, empirical studies validate the effectiveness of the proposed method.

**Questions:**

Typos:
1. Line 179: convenient provably convenient -> provably convenient

**Ethics Review Area:**

["I don’t know"]

**Limitations:**

Not much.

**Strengths And Weaknesses:**

Strengths:
1. This work gives a solution to this novel problem along with the corresponding upper bound, which also closely mimics the behavior of the lower bound.
2. The overall paper is well-written and quite readable, especially some discussions about the intuition behind certain conclusions.

Weakness:
Not much.

---

> ### Author Response · Authors · 2022-08-01
> **Thank the reviewer - typo fixed**
>
> We thank the reviewer for taking the necessary time and effort to review our paper. We are happy that the reviewer appreciated our work. We fixed the typo in the version that we recently uploaded.

---

### Official Review · Reviewer_jriY · 2022-07-12

**Rating:** 8
**Confidence:** 4
**Soundness:** 4 excellent
**Presentation:** 3 good
**Contribution:** 3 good

**Summary:**

The paper studies multi-fidelity best arm identification (BAI). Here, we have a bandit model
with M different fidelities. On each round, a decision-maker should choose which arm to
pull and which fidelity to pull the arm at. It's goal is to identify the best arm (the arm
with the largest mean at the highest fidelity), but can pull the arms at lower fidelities
to obtain cheaper approximations at lower cost. The authors assume that the lower
fidelities may be biased from the highest fidelity, but these biases are bounded by some known
quantity.
They first prove a lower bound for this problem and describe a racing-style algorithm
which achieves this lower bound under certain assumptions on the biases and costs.


Multi-fidelity bandits are well-studied in the regret minimization setting under the same
model considered by the authors
However, I am not
aware of papers which have studied multi-fidelity BAI. It is a very natural formalism with
useful applications.
While the paper falls short of solving this problem completely, I think the authors have
made meaningful advances.
Hence, I believe it would be a nice addition to the proceedings.

**Questions:**

See above

**Limitations:**

See above

**Strengths And Weaknesses:**

The lower bound given in the paper is quantifiably better than the lower bound for regular
BAI. However, I am not sure if it is tight enough. In particular, taking the minimum on
each term in the summation seems a bit too optimistic as it doesn't account for the
exploration cost at other fidelities.

Related to the above: Assumption 1 seems to only consider problems where there is large
separation between fidelities (i.e. the costs increase sharply between a lower fidelity to
a higher while the gaps decrease sharply). This rules out many challenging problems where
the fidelities might be similar (i.e. similar costs/gaps); here, you might still expect a
smart enough policy to use that information or quickly ignore that information and move to
higher fidelities.

It might be useful to discuss the above issues to elucidate on the gap between the lower
and upper bounds. My hunch is that they will both need to be tightened outside of
Assumption 1.

Other comments:
Footnote 1 is very unclear, what does it mean for costs and fidelity to hold with
equality and what does it mean for it to be discarded. You need to be more clear and
precise about what is meant here.

---

> ### Author Response · Authors · 2022-08-01
> **On closing the theoretical problem**
>
> We thank the reviewer for the effort in reading our work, for the positive comments, and for the in-depth discussion that has been pointed out.
>
> We agree with the reviewer that both our upper and lower bounds could be improved to close the problem from a theoretical perspective.
> More specifically, we agree that the lower bound seems currently optimistic since it does not encode the cost of exploring other fidelities.
>
> To tackle this issue from a lower bound perspective, we notice that proof techniques that rely on the characteristic time function (i.e., "Optimal Best Arm Identification with Fixed Confidence", Garivier et al., 2016) might lead to tighter bounds. Consequently, adapting to our multi-fidelity (MF) setting their well-known "Track and Stop" strategy might provide upper bounds that asymptotically match the lower bound in its characteristic time form. This represents an exciting line for future research. However, We want to remark that results obtainable from these approaches usually come at the cost of lower interpretability (and, depending on the nature of the resulting characteristic function, computational complexity). On the other hand, our work is the first that addresses best-arm identification (BAI) with MF and already provides several insights into the challenges and key properties of exploiting biased approximators in BAI problems. Furthermore, our algorithm is computationally efficient.
>
> The previous comment is also clearly related to Assumption 1. As the reviewer suggests, outside the setting proposed in Assumption 1, the cost of "exploring" lower fidelities becomes more and more relevant. Indeed, Assumption 1 guarantees that there is a large enough "fidelity gap" (i.e., fidelity differs in terms of the joint costs-biases relationship specified by the assumption). The main issue in settings where this gap is small roots down to the chicken-egg problem of MF-BAI. On one side, we would like to exploit these approximators to discard sub-optimal arms; on the other side, we do not know if these approximators will be useful (i.e., in the worst case, we pay an exploration cost). In the case where the "fidelity-gap" is small, this problem is repeated over and over for all these fidelities close to one another (i.e., the cumulative sum of the exploration costs becomes non-negligible). In this sense, tighter lower bounds (such as the ones previously suggested) might shed light on whether such assumptions are necessary or not, or if tighter assumptions are possible/necessary. Future works should definitely aim at tackling these sorts of problems.
>
> Concerning Footnote 1, we meant that settings in which we have $\lambda_i = \lambda_{i+1}$ or $\xi_i = \xi_{i+1}$ for some $i$ (i.e., the footnoted relationship holds with equality) could be easily ruled out by a pre-processing algorithm that runs at the beginning of the learning process, and that eliminates from the agent knowledge fidelity which is dominated by others. In simpler words, we run the algorithm without considering these fidelities (i.e., dominated fidelities are discarded). Consider to have $\lambda_1 = 1$, $\lambda_2 = 1000$, $\xi_1 = 0.1$, $\xi_2 = 0.1$, then, we can simply get rid of fidelity $m=2$, since, for the same accuracy level, we can get cheaper samples from $m=1$. We have clarified this concept in the new version we have uploaded.

---

> > ### Comment · Reviewer_jriY · 2022-08-08
> > **Increasing score**
> >
> > I thank the authors for the response which has addressed some of the concerns.  I have increased my score to 8. I encourage the authors to discuss the limitations of Assumption 1 in the paper.

---

> > > ### Author Response · Authors · 2022-08-08
> > > **Document update**
> > >
> > > We thank again the reviewer for the discussion. We have currently inserted the discussion on Assumption 1 in the appendix. We will make use of the additional space to include it in the main text for the camera-ready version.
> > >
> > > We thank the reviewer for increasing the score to 8. We will kindly ask the reviewer to check if the procedure was successful since we still see the old score.

---

### Official Review · Reviewer_9j2c · 2022-07-15

**Rating:** 5
**Confidence:** 3
**Soundness:** 3 good
**Presentation:** 3 good
**Contribution:** 2 fair

**Summary:**

This paper presents a new algorithm and analysis for multi-fidelity
best-arm identification.  It builds on the ideas of previous
algorithms that do successive elimination of arms until only one is
left.  A unique aspect is their joint consideration of the costs and
accuracies of each fidelity level.  They demonstrate their method
empirically on a couple toy/synthetic problems.


**Questions:**

none

**Strengths And Weaknesses:**

The paper is well written and covers the prior work well.  My only
concern is that the contributions are relatively incremental given the
large amount of prior work on best arm identification, and
multi-fidelity methods.

The idea of using multi-fidelity best arm identification as a planning algorithm is interesting.  Perhaps that needs to be left for another paper but it would be interesting to explore it more and see how it compares to SOTA planning algorithms.

---

> ### Author Response · Authors · 2022-08-01
> **Comparison with related works**
>
> We thank the reviewer for the effort in carefully reading our work and for the valuable comments. As the reviewer suggests, there is a large amount of prior work on best-arm identification (BAI) and multi-fidelity (MF) methods. However, we would like the reviewer to ponder the following aspects:
>
> * **No prior work** has tackled the challenges that arise in the joint MF-BAI setting (i.e., best-arm identification with multi-fidelity). This requires devising a novel notion of performance index. The **cost complexity** index introduced in our paper is novel and significantly differs from pseudo-regret and simple regret metrics studied in the literature.
>
> * Almost all prior works in the MF setting (except for "The multi-fidelity multi-armed bandit" Kandasamy et al., 2016) make **structural assumption** on the function that needs to be studied (i.e., the arm space). As one can read in Section 6, smoothness conditions implied by Gaussian Processes (GP) or the hierarchical partition assumption on the target function are usually employed. In our work, such assumptions are not present. In this sense, in the following discussion, we refer to our setting as the multi-fidelity "unstructured best-arm identification" problem.
>
> These two aspects have several implications for the design of the algorithm and its theoretical analysis. More specifically, the unstructured arm-space setting in the best-arm identification domain allows us to focus on the following key questions: *"To what extent can we exploit biased approximators in unstructured best-arm identification problems?"*. In this sense, Section 3 provides an insightful lower bound on the inner structure of the problem when no additional assumption on the arm space is available (and our algorithm in Section 4 reflects this peculiar structure). When a smoothness/hierarchical assumption is present, these aspects are strictly dependent on the assumption itself, with a significant impact both on the theoretical analysis and the algorithmic design. For instance, consider having a GP kernel on the joint arm-fidelity space (e.g., "Multi-fidelity Bayesian Optimisation with Continuous Approximations", Kandasamy 2017). In this case, querying a given point $x$ at the most imprecise fidelity provides information on everything else, which is significantly stronger than what we consider in our setting and hides the true nature of the key question that we aim to answer.
>
> Furthermore, even in the case of finite arm-space with no assumption ("The multi-fidelity multi-armed bandit" Kandasamy et al., 2016), the difference in the performance index requires ad-hoc choices and theoretical tools (see, for instance, the Fidelity Convenience, Definition 1, or the proof of the lower bound).
> Finally, we remark that for the finite unstructured arm spaces our work sheds light on:
>
> * Assumption that makes the multi-fidelity structure provably convenient. Our assumption directly encodes relationship between costs and biases.
>
> * How to exploit **order-aware knowledge**. This is something that we expect from a MF method, and whose impact in the finite-arm case was, up to now and to the best of our knowledge, unknown. The formulation that we propose for order-aware knowledge (i.e., $\gamma$): a) has intuitive meaning, b) leads to performance improvement, both theoretically and empirically, and c) has practical relevance since the formulation we propose is available in a large number of domains.
>
> To conclude, we agree that using multi-fidelity best-arm identification techniques for planning problems deserves attention and should be analyzed in more detail. We are currently developing this idea in greater detail for future works.
>
>
> We hope that these comments clarify the doubts the reviewer raised. We will be glad to answer further concerns on this topic.

---

> > ### Author Response · Authors · 2022-08-06
> > **Follow up**
> >
> > We wanted to follow up on the response to the previous comments. In particular, are the reviewer’s doubts regarding comparison with related works solved?
> > If yes to the above, is the reviewer satisfied with the overall response? If no, would the reviewer be willing to engage in further discussion about the disagreements?
> > Thanks again for your effort and comments!

---

### Meta-Review · Area_Chair_C1sg · 2022-08-27

**Recommendation:** Accept
**Confidence:** Certain

**Metareview:**

This paper considers the multi-fidelity variant of the best-arm identification problem. I recommend its acceptance and I strongly encourage the authors to take the several fantastic points raised by the reviewers while crafting their next draft. For instance, please include a discussion about (and clarification of) Assumption 1 that reflects the discussion with the reviewers.

**Award:**

No

---

### Decision · Program_Chairs · 2022-09-14

Accept